An enigmatic aquatic snake from the Cenomanian of Northern South America

Albino Adriana 1 aalbino@mdp.edu.ar
Carrillo-Briceño Jorge D. 2
Neenan James M. 3 4
1 Departamento de Biología, Universidad Nacional de Mar del Plata-CONICET , Mar del Plata , Argentina
2 Paläontologisches Institut und Museum, Universität Zürich , Zürich , Switzerland
3 Oxford University Museum of Natural History , Oxford , United Kingdom
4 Department of Earth Sciences, University of Oxford , Oxford , United Kingdom
Sues Hans-Dieter
Electronic publication date: 2016 May 24
Publication date: 2016
Volume: 4
Electronic Location ID: e2027
Received 2016 Jan 27; Accepted 2016 Apr 19
Copyright: ©2016 Albino et al.
Copyright year: 2016
Copyright holder: Albino et al.
License: This is an open access article distributed under the terms of the Creative Commons Attribution License, which permits unrestricted use, distribution, reproduction and adaptation in any medium and for any purpose provided that it is properly attributed. For attribution, the original author(s), title, publication source (PeerJ) and either DOI or URL of the article must be cited.
License URL: https://creativecommons.org/licenses/by/4.0/

Keywords: South America, Cretaceous, Venezuela, Snakes, La Luna Formation

Funding: PIP-CONICET 112-200901-00176 Swiss National Science Foundation 31003A-149605 P2ZHP3_162102 This project was supported by the PIP-CONICET No 112-200901-00176 (AA), and Swiss National Science Foundation grants 31003A-149605 (awarded to James M. Neenan) and P2ZHP3_162102 (JMN). The funders had no role in study design, data collection and analysis, decision to publish, or preparation of the manuscript.

==============================
We report the first record of a snake from the Cretaceous of northern South America. The remains come from the La Luna Formation (La Aguada Member, Cenomanian of Venezuela) and consist of several vertebrae, which belong to the precloacal region of the vertebral column. Comparisons to extant and extinct snakes show that the remains represent a new taxon, Lunaophis aquaticus gen. et sp nov. An aquatic mode of life is supported by the ventral position of the ribs, indicating a laterally compressed body. The systematic relationships of this new taxon are difficult to determine due to the scarcity of fossil material; it is, however, a representative of an early lineage of snakes that exploited tropical marine pelagic environments, as reflected by the depositional conditions of the La Aguada Member. Lunaophis is also the first aquatic snake from the Cenomanian found outside of the African and European Tethyan and Boreal Zones.

Introduction

Until recently, the oldest record of snakes has been from the Albian of Algeria (Cuny et al., 1990) and the Albian–Cenomanian of North America (Gardner & Cifelli, 1999), whereas a supposed snake from the Barremian of Spain (Rage & Richter, 1994) was recently excluded from the group (Rage & Escuillié, 2003). However, these records have few phylogenetically informative characters and add little to the knowledge of the origin and early evolution of snakes.

Using a combination of traditional node-based dating and novel fossil tip-dating methods, Hsiang et al. (2015) determined that the snake total-group originated during the middle Early Cretaceous (∼128.5 Ma), with the crown-group following about 20 million years later, during the Albian stage. Nevertheless, new studies on squamate specimens from the Jurassic (Bathonian and Kimmeridgian), which include cranial and postcranial remains of at least four different species from the USA (Diablophis gilmorei Caldwell et al., 2015), Portugal (Portugalophis lignites Caldwell et al., 2015), and England (Parviraptor estesi Caldwell et al., 2015 and Eophis underwoodi Caldwell et al., 2015), suggest that snakes have undergone habitat differentiation and evolutionary radiation at least since the mid-Jurassic (Caldwell et al., 2015).

In South America, the oldest snakes are known from the Mesozoic of Brazil and Argentina. The Brazilian taxa consist of the putative four-limbed snake Tetrapodophis amplectus Martill, Tischlinger & Longrich, 2015 from the Early Cretaceous (Aptian), and the small Seismophis septentrionalis Hsiou et al., 2014 from the Late Cretaceous (Cenomanian). The fossil record from Argentina represents one of the richest fauna of early snakes, including the hind-limbed Najash rionegrina Apesteguía & Zaher, 2006, from the Cenomanian (Apesteguía & Zaher, 2006; Zaher, Apesteguía & Scanferla, 2009; Palci, Caldwell & Albino, 2013), the medium-sized snake Dinilysia patagonica Smith-Woodward, 1901, from the Santonian–Campanian (Smith-Woodward, 1901; Estes, Frazzetta & Williams, 1970; Hecht, 1982; Rage & Albino, 1989; Caldwell & Albino, 2002; Caldwell & Calvo, 2008; Zaher & Scanferla, 2012), the small “anilioid” Australophis anilioides Gómez, Báez & Rougier, 2008, and diverse taxa of Madtsoiidae from the Campanian–Maastrichtian (Albino, 1986; Albino, 1994; Albino, 2000; Albino, 2007; Albino, 2011a; Albino, 2011b; Martinelli & Forasiepi, 2004). In this paper we describe a new taxon of aquatic snake based on vertebrae from the Cenomanian La Luna Formation in the Andes of Venezuela. This specimen represents the oldest known record of snakes from northern South America and adds substantial information about the diversity of the group during its evolution.

Figure 1 Location map of the Cementos Andinos quarry, Trujillo estate, Venezuela.

Geological setting

The specimen was found in strata of the La Luna Formation (La Aguada Member), exposed in a cement quarry (Cementos Andinos Company) in the Andes of Venezuela, east of Lake Maracaibo, 10 km northeast of Monay in the Candelaria Municipality of Trujillo State (Fig. 1).

The Upper Cretaceous La Luna Formation is the most prolific petroleum source rock in western Venezuela and part of eastern Colombia (Zumberge, 1984; Tribovillard et al., 1991; Zapata et al., 2003), and is characterized by a sequence of marine rocks deposited under anoxic-poorly oxygenated conditions along the passive margin of northern South America during the Cenomanian to Campanian (Zapata et al., 2003). This lithostratigraphic unit was originally named the ‘La Luna Limestone’ by Garner (1926) in the Quebrada La Luna of the Perijá range (Zulia state, western Venezuela), being formally described as a formation by Hedberg & Sass (1937). The lithology of the La Luna Formation is characterized by alternating black or dark-gray limestones and organic calcareous shales, in which the calcareous concretions are abundant (González de Juana, Iturralde de Arocena & Picard, 1980; Tribovillard et al., 1991; Davis, Pratt & Sliter, 1999). Renz (1959) subdivided the La Luna Formation into three members that are exposed in the southeast of the Maracaibo basin in the Lara and Trujillo states: the lower La Aguada Member (∼60 m thick of dense, black/dark-gray limestones and black or brown shales), the middle Chejendé Member (∼80 m thick of black shales and marls), and the upper Timbetes Member (∼90 m thick of laminated limestones and shales) (Fig. 2A). Siliceous and phosphatic horizons characterize the top of the unit, recognizing the Ftanite of Táchira (Coniacian-Santonian) and Tres Esquinas members (upper Campanian), respectively. The Tres Esquinas Member is well exposed in the Cordillera de Mérida and Perijá whereas the Ftanite of Táchira Member is exposed mainly at the southwest of the Cordillera de Mérida in Táchira State (González de Juana, Iturralde de Arocena & Picard, 1980; De Romero & Galea, 1995; Erlich et al., 2000).

Figure 2 Stratigraphic context.

(A) Cretaceous lithostratigraphic units of the Chejendé region, near Monay city, Trujillo State (modified after Renz, 1959 and González de Juana, Iturralde de Arocena & Picard, 1980); (B) Stratigraphic section of the Aguada Member in the Cementos Andinos quarry.

The outcrops of the Aguada Member in the Cementos Andinos quarry (Figs. 2B and 3A, 3B) present a succession of dense dark-gray limestones of up to ∼60–70 cm thick, intercalated with laminated black, dark-gray or brown shales. Molluscs, fish remains, and hard, discoidal or ellipsoidal calcareous concretions, reaching up to 198 cm in diameter (Figs. 3C and 3D), are common throughout the section (Fig. 2B). In the studied section, the strata are inclined almost vertically (Fig. 3A), and its base overlays a fossiliferous dark-gray sandy limestone, which has been recognized in the Andes of Trujillo and Lara as the top of the upper Albian Maraca Formation (González de Juana, Iturralde de Arocena & Picard, 1980). Nevertheless, other authors (Renz, 1968; Erlich et al., 1999) have used the name of La Puya Member to refer to a thin section (<30 m) at the top of the Peñas Altas Formation in the Andes of Lara and Trujillo. Therefore, the discrepancy between the use of Maraca Formation or La Puya Member for the thin sequence underlying the Aguada Member is still unresolved. A Cenomanian age for the La Aguada Member has been provided by planktonic foraminiferans and ammonites (Renz, 1959).

Figure 3 Outcrops of the La Aguada Member in the Cementos Andinos quarry.

(A) Fossiliferous strata; (B) Strata with calcareous concretions; (C), (D). Discoidal calcareous concretions.

Materials and Methods

The studied specimen (MNCN-1827) is deposited in the Museo de Ciencias Naturales de Caracas, Venezuela. The fossil (Figs. 4–6) was compared directly with osteological material from a diverse group of present-day squamates in the Colección Herpetológica de la Universidad Nacional de Mar del Plata- Sección Osteología, Argentina (UNMdP-O). Measurements were taken with manual calipers.

Figure 4 Holotype of Lunaophis aquaticus.

MNCN-1827-A, isolated precloacal vertebra. Anterior (A), left lateral (B), dorsal (C), and ventral (D) views.

Figure 5 Holotype of Lunaophis aquaticus, isolated precloacal vertebrae.

(A–B), MNCN-1827-B; (C–D), MNCN-1827-C; (E–F), MNCN-1827-D; (G–H), MNCN-1827-E. Dorsal (A, C, E, G), and ventral (B, D, F, H) views.

Figure 6 Holotype of Lunaophis aquaticus.

MNCN-1827-F, articulated precloacal vertebrae. Dorsal (A) and ventral (B) views.

Following the criteria of Rage, Vullo & Néraudeau (2016), pachyostosis s.l. actually corresponds to two processes: pachyostosis s.s. and osteosclerosis. The first is considered as the hyperplasy of periosteal cortices; it entails modifications of the morphology and its presence is therefore directly observable. Osteosclerosis corresponds to an increase of the compactness of the inner bone; it does not affect morphology and it can be seen only in bone sections.

The specimen was scanned using micro-computed tomography (μCT) with a Scanco Medical μCT80 machine at the Anthropological Institute, University of Zurich, Switzerland. The specimen was scanned using a voltage of 70 kV and an intensity of 114 µA, resulting in a slice thickness/increment of 18 µm. The resulting slice data were then processed and 3D models created using Avizo 8, revealing inner structures of the vertebrae such as thickened cortical bone.

Phylogenetic relationships of fossil and Recent snakes proposed by Hsiang et al. (2015) are adopted in this paper. These authors present the most complete picture of the early evolution of snakes to date, including fossils, phenotypic and genetic data in combined phylogenetic analyses to reconstruct the ancestor of the snake total-group and of crown snakes, using both established and recently developed analytical methodologies.

The electronic version of this article in Portable Document Format (PDF) will represent a published work according to the International Commission on Zoological Nomenclature (ICZN), and hence the new names contained in the electronic version are effectively published under that Code from the electronic edition alone. This published work and the nomenclatural acts it contains have been registered in ZooBank, the online registration system for the ICZN. The ZooBank LSIDs (Life Science Identifiers) can be resolved and the associated information viewed through any standard web browser by appending the LSID to the prefix http://zoobank.org/. The LSID for this publication is: urn:lsid:zoobank.org:pub:918B6879-8908-488F-876B-EA741DFF627B. The online version of this work is archived and available from the following digital repositories: PeerJ, PubMed Central and CLOCKSS.

Results

Systematic paleontology

Squamata Oppel, 1811	
Serpentes Linnaeus, 1758	
Lunaophis aquaticus, gen. et sp. nov. urn:lsid:zoobank.org:act:175D3D55-D85A-4013-8D30-563BAB7A4143	
Figs. 4–10	

Holotype. MNCN-1827. The type specimen is composed of vertebral remains in a small block of black shale, which belong to a single individual. All vertebrae are consistent in size. The remains include: an almost complete isolated preclocal vertebra (MNCN-1827-A, Fig. 7), an isolated precloacal vertebra that lacks the left prezygapophysis (MNCN-1827-B, Fig. 8), two isolated and incomplete precloacal vertebrae (MNCN-1827-C, Figs. 9A–9D, and MNCN-1827-D, Figs. 9E–9H), an isolated and partially preserved precloacal vertebra (MNCN-1827-E, Figs. 9I–9L), five articulated precloacal vertebrae (MNCN-1827-F, Figs. 10A–10D), and a poorly preserved vertebral fragment (MNCN-1827-G, Figs. 10E–10H).

Figure 7 Holotype of Lunaophis aquaticus.

MNCN-1827-A, isolated precloacal vertebra. Anterior (A), posterior (B), dorsal (C), ventral (D), right lateral (E), left lateral (F), dorsoposterior (G), and posterolateral (H) views; co, condyle; ct, cotyle; hk, haemal keel; izr, interzygapophyseal ridge; na, neural arch; pd, paradiapophysis; plf, paralymphatic fossa; po, postzygapophysis; pr, prezygapophysis; sbr, subcentral ridge; zg, zygosphene.

Figure 8 Holotype of Lunaophis aquaticus.

MNCN-1827-B, isolated precloacal vertebra. Dorsal (A), ventral (B), left lateral (C), and right lateral (D) views; co, condyle; hk, haemal keel; na, neural arch; ns, neural spine; pd, paradiapophysis; plf, paralymphatic fossa; po, postzygapophysis; pr, prezygapophysis; sbr, subcentral ridge; zg, zygosphene.

Figure 9 Holotype of Lunaophis aquaticus.

(A–D) MNCN-1827-C, isolated precloacal vertebra; (E–H) MNCN-1827-D, isolated precloacal vertebra; (I–L), MNCN-1827-E, isolated anterior vertebra. Dorsal (A, E, I), ventral (B, F, J), left lateral (C, G, K), and right lateral (D, H, L) views; co, condyle; na, neural arch; ns, neural spine; pd, paradiapophysis; po, postzygapophysis; pr, prezygapophysis; sbr, subcentral ridge.

Type locality and horizon. Cement quarry (Cementos Andinos company), located east of Lake Maracaibo, 10 km northeast of Monay, Trujillo State, Venezuela (Fig. 1). The fossiliferous horizon is a black shale layer ∼28 m above the base of the La Aguada Member of the La Luna Formation (Cenomanian, Renz, 1959, Fig. 2).

Etymology. Lunaophis: snake from La Luna, denotes the origin of the material from rocks corresponding to La Luna Formation; Latin aquaticus: water-dwelling.

Diagnosis. Medizum-sized precloacal vertebrae bearing neural arch wide, depressed, and longer than the vertebral centrum, with a convex posterior edge covering the view of the condyle in dorsal view. Lateral walls of the arch arising from the midline from where they diverge ventrally to the subcentral ridges. Body of the prezygapophyses swollen and neural arch above postzygapophyses constituting protruded bulges (pachyostosis). High development of subcentral and interzygapophyseal ridges. Long and narrow vertebral centrum bearing a low and anterior placement of small paradiapophyses. Paradiapophyses closely spaced and strongly projected ventrally beyond the ventral rim of the cotyle through a short process. Articular surfaces of para- and diapophyses facing ventrally. Neural spine well-developed only along the anterior part of the precloacal region of the column as a posterior tubular process projecting free beyond the posterior edge of the neural arch. Neural spine drastically reduced in mid-trunk vertebrae, and finally lost, in posterior trunk vertebrae.

Description. In general, the vertebrae are medium-sized (Table 1), comparable to mid-trunk vertebrae of specimens of the extant boid Epicrates alvarezi that were around 1.5 m long in life (AA, pers. obs., 2015 on specimens from the UNMdP-O collection). They are low (H), long (nal, pr-po) and wide (pr-pr, po-po). The compared dimensions demonstrate that they are wider than high (pr-pr or po-po > H). Two of the best-preserved vertebrae are MNCN-1827-A (Figs. 4 and 7) and MNCN-1827-B (Figs. 5 and 8). Vertebra MNCN-1827-B is slightly smaller than vertebra MNCN-1827-A, but the general aspect and characters are the same, except for slight differences. The following comprehensive description is based on these vertebrae, although diagnostic characters are true for all.

Table 1 Measures available on vertebrae of Lunaophis aquaticus.

Measures (in mm)	MNCN-1827-A	MNCN-1827-B	MNCN-1827-C	MNCN-1827-D	
cl	8.44	8.00	7.62	–	
cow	3.00	2.86	3.00	–	
cth	–	2.82	3.00	2.58	
ctw	–	3.10	3.18	3.00	
H	6.34	5.70	7.16	–	
naw	6.44	6.18	6.34	6.84	
nal	10.16	9.22	–	–	
po-po	–	9.62	–	–	
prl	3.56	3.34	3.50	3.26	
prw	2.56	2.76	2.40	2.84	
pr-po	9.52	–	9.20	9.14	
pr-pr	10.00	–	–	10.94	
zgh	–	0.40	–	–	
zgw	3.66	3.60	–	–	
Notes.

Abbreviationscl centrum length

cow condyle wide

cth cotyle high

ctw cotyle wide

H high of the vertebra

naw neural arch wide

nal neural arch length

po-po distance between postzygapophyses

prl prezygapophyses length

prw prezygapophysis wide

pr-po distance between pre- and postzygapophyses of the same side

pr-pr distance between prezygapophyses

zgh zygosphene high

zgw zygosphene wide

In anterior view, the zygosphene is well-developed and slightly wider than the cotyle (zgw > ctw); it is thin in the middle and its dorsal edge is almost flat. The articular facets are relatively large and anteriorly oriented. The neural canal is small, with a round outline. The prezygapophyses are robust, inflated and large; they are borne at the ventral base of the neural canal and slant above the horizontal plane, but do not reach the level of the zygosphenal roof. There are no prezygapophyseal processes. The cotyle is large, nearly circular, and delimited by a well-marked rim. It is partially filled by sediment. There are strong depressions on both sides of the cotyle but the paracotylar foramen is visible only on the right side of vertebra MNCN-1827-A. In specimen MNCN-1827-B there is not a visible paracotylar foramen on the right side and it is broken on the left. The paradiapophyses are positioned ventral to the cotyle, distant from the prezygapophyseal surfaces and close to each other; they project ventrally with a short and constricted process separating them from the vertebral centrum. The articular surfaces are small, with clearly distinctive parapophyses and diapophyses facing ventrally and strongly projecting beyond the ventral rim of the cotyle.

In posterior view, the neural arch is depressed and forms a protruded bulge above the postzygapophysis as a strong, inflated convexity, especially on vertebra MNCN-1827-B on both sides and on vertebra MNCN-1827-A on the left. These swollen paired posterior portions of the neural arch platform are interpreted as pachyostosis. The zygantra filled by sediment, which also extends over the dorsal condyle. The roof of the neural arch of vertebra MNCN-1827-B is proportionally less depressed than in specimen MNCN-1827-A, and the zygantra are larger. Vertebra MNCN-1827-A has the left postzygapophysis distally broken whereas in vertebra MNCN-1827-B the fracture is on the right. The postzygapophyseal surfaces are large and slightly inclined above the horizontal. There are no parazygantral foramina. The condyle is large and more or less round. The posterior end of a wide hemal keel is slightly visible ventral to the condyle.

In dorsal view, the neural arch is long and wide, with the posterior edge convex in vertebra MNCN-1827-A and almost straight in vertebra MNCN-1827-B. None of the condyle is visible in this view. The interzygapophyseal constriction is concave but not especially deep. The interzygapophyseal ridges are strongly developed and protrude laterally beyond the level of the lateral walls and subcentral ridges. They connect the pre- and postzygapophysis of the same side. The articular surfaces of the prezygapophyses are large, oval, longer than wide, and anterolaterally oriented. The zygosphene is well-developed and concave in the middle. It is partially broken on the left side of specimen MNCN-1827-A whereas it is complete on vertebra MNCN-1827-B. Vertebra MNCN-1827-A does not have neural spine; only a minor crest restricted to the anteriormost part of the neural arch. It is limited by narrow and short depressions at each side. Posteriorly to it, the surface of the arch is smooth. In contrast, a well-defined but very low neural spine is developed, but partially broken in the middle, along most of the roof of the neural arch in vertebra MNCN-1827-B. Longitudinal and marked ridges are extended anteroposteriorly on either side of the neural spine; they are laterally flanked at both sides by deep grooves. These crests and grooves are prominent especially in the posterior half, reaching the posterior border of the neural arch. Distally, over each postzygapophysis, the neural arch forms protruded, swollen bulges representing pachyostosis. The bulging is not developed on the right side of vertebra MNCN-1827-A producing asymmetry.

In ventral view, the vertebral centrum is long (cl > naw) and narrow, slightly wider anteriorly than posteriorly, but not markedly triangular in section. The subcentral ridges are well defined and prominent (Figs. 7–9) and the paralymphatic fossae are well-developed (Figs. 7 and 8). The cotyle is almost not exposed ventrally whereas the condyle is well exposed from this view. The ventral surface of the centrum is concave, with a distinctive but weakly developed rounded hemal keel, which is smooth anteriorly and more defined and wider posteriorly. The short precondylar constriction is marked. There no subcentral foramina. The paradiapophyses are small with articular surfaces well exposed ventrally. The di- and parapophyseal surfaces are distinct and separated by a short and deep constriction.

In lateral view, the vertebrae are long, with significantly depressed neural arch roofs. Anteriorly, the neural arch extends beyond the level of the cotyle due to the anterior projection of the zygosphene. Posteriorly, the neural arch is longer than the vertebral centrum (nal > cl), extending beyond the level of the condyle. The neural arch in the vertebra MNCN-1827-B is slightly shorter than in specimen MNCN-1827-A. The neural spine is partially broken in vertebra MNCN-1827-B but it would have been long and low, as a thin crest developed from the base of the zygosphene and almost until the posterior end of the neural arch. The zygosphenal surfaces are prominent, oval, longer than wide, and more anteriorly than dorsally oriented. Posteriorly, on either side, the roof of the neural arch are inflated forming the mentioned hemispheric bulge above the postzygapohysis. As a result, the outline of the neural arch is concave in lateral view. The absence of a neural spine in vertebra MNCN-1827-A produces a more deeply concave arch in lateral view than in vertebra MNCN-1827-B. The prezygapophyses are large, swollen, and anterolaterally oriented. The interzygapophyseal crest is well marked, laterally projected, and strongly separates the roof from the lateral walls of the neural arch. The distance between the interzygapophyseal crests (naw) is much higher than the distance between the lateral walls of the neural arch where they contact with the roof. This is because the lateral walls are borne near the saggittal axis of the vertebra. They diverge dorsoventrally from this point to the subcentral ridges. This structure produces a prominent shelf-like arch roof of the neural arch on each side between the pre- and postzygapophysis (Fig. 7H). There are no lateral foramina. The vertebral centrum is long but shorter than the neural arch (nal > cl). The subcentral ridges are prominent. The main axis of the condyle is not strongly inclined from the horizontal plane. The paradiapophyses are low, and clearly separated from the centrum by a deep constriction at the end of a short projection. They are small and ventrally extend beyond the ventral edge of the condyle and subcentral ridges.

Vertebra MNCN-1827-C (Figs. 5 and 9) is approximately the same size as vertebra MNCN-1827-B. The zygosphene, left prezygapophysis, and part of the posterior part of the neural arch are not preserved. This vertebra is slightly deformed. There is no neural spine, as in vertebra MNCN-1827-A.

Vertebra MNCN-1827-D (Figs. 5 and 9) is also similar in size to specimen MNCN-1827-B. It has lost most of the vertebral centrum, right postzygapophysis, and the zygosphene. In dorsal view, the posterior edge of the neural arch is strongly convex medially, more than in vertebra MNCN-1827-A.

Specimen MNCN-1827-E is a poorly preserved vertebra (Figs. 5 and 9). The entire left side as well as the cotyle and zygosphene are missing. This vertebra is also slightly deformed and has deposits of sediment. Its size is similar to that of vertebra MNCN-1827-B. It also has the same general characters except that the anteriorly low neural spine rises abruptly at the distal part of the neural arch, becoming a well-developed neural spine which is posteriorly directed in a strong angle and projects freely beyond the posterior edge of the neural arch (Figs. 9I–9L). The free portion of the spine comprises approximately 63% of the length of the neural arch and thus forms more than half the total length of the vertebra. It has a tubular form; it is thin and its anteroposterior section is short. This spine forms an acute angle with respect to the roof of the neural arch (less than 45°), therefore not increasing significantly the height of the vertebra. The vertebra does not have a hypapophysis on the ventral surface of its vertebral centrum. The most posterior part of the centrum bears a slight prominence not wider than the cotyle, similar to the posterior part of a hemal keel in specimen MNCN-1827-B. Based on the presence of a well-developed neural spine, this vertebra is interpreted as an anterior trunk vertebra (see ‘Discussion’).

Specimen MNCN-1827-F includes five tightly articulated vertebrae (Figs. 6 and 10), with the same morphology as vertebra MNCN-1827-A and without any trace of neural spines. Based on the µCT sections of these vertebrae, both the centrum and the neural arch are revealed to be osteosclerotic, as they are composed of thick cortical bone (Fig. 11). Fragment MNCN-1827-G does not have distinctive features (Figs. 10E–10H).

Figure 10 Holotype of Lunaophis aquaticus.

(A–D), MNCN-1827-F, articulated precloacal vertebrae; (E–H), MNCN-1827-G, scarcely preserved vertebral fragment. Dorsal (A, E,), ventral (B, F), right lateral (C, G), and left lateral (D, H) views; hk, haemal keel; na, neural arch; ns, neural spine; pd, paradiapophysis; po, postzygapophysis; pr, prezygapophyis.

Discussion

Variations along the vertebral column

Following the atlas and axis, the vertebral column of snakes is conventionally divided into precloacal (trunk), cloacal and caudal regions (Hoffstetter & Gasc, 1969). Cloacal and caudal vertebrae are characterized by smaller size and the presence of additional apophyses which are absent in all vertebrae of the studied sample. Nevertheless, some vertebrae of Lunaophis lack any trace of neural spine (MNCN-1827-A, MNCN-1827-C, MNCN-1827-F), others have a very low neural spine, present as a thin and anteroposteriorly elongate crest (MNCN-1827-B and probably MNCN-1827-D), and finally one has a well-developed neural spine projecting from the neural arch roof (MNCN-1827-E). Hence, the diverse neural spines in different vertebrae of the same individual of Lunaophis imply significant variation along the precloacal region of the vertebral column. In snakes, the precloacal region is subdivided into three portions, the limits of which are imprecisely defined: the anterior, middle and posterior regions (Hoffstetter & Gasc, 1969; Albino, 2011a; Albino, 2011b). The similar size of the studied vertebrae of Lunaophis indicates that they are not anteriormost or posteriormost precloacal vertebrae, which should be smaller (as is the case in all snakes). In addition, anteriormost precloacal vertebrae of snakes are characterized by the presence of hypapophyses on the ventral surface of the vertebral centra at least until the fourth vertebra (Hoffstetter & Gasc, 1969), whereas in all preserved vertebrae of Lunaophis, there is a hemal keel instead of a hypapophysis below the vertebral centrum. No other significant differences occur among the studied vertebrae of the sample, except for the development of the neural spine. Taking into account that neural spines in other snakes are well-developed in the anterior precloacal region but become reduced poteriorly (Hoffstetter & Gasc, 1969; Albino, 2011a; Albino, 2011b), the vertebra with a well-developed neural spine of Lunaophis would be an anterior vertebra (probably near the limit with the mid-trunk region). They would be followed by vertebrae with very low neural spines (mid-trunk vertebrae) and then by posterior trunk vertebrae that lack neural spines (probably near the limit with the mid-trunk region). Thus, neural spines in Lunaophis would have been well-developed only along the anterior part of the vertebral column, including some vertebrae that have already lost the hypapophyses. They become drastically reduced, and finally lost, posteriorly.

Figure 11 Longitudinal (A) and transverse (B) μCT sections through specimen MNCN-1827-F showing pachyostotic, thickened cortical bone; with section positions shown in C.

Abbreviations: tcb, thickened cortical bone; nc, neural canal.

Comparative osteology

The overall morphology of the vertebrae in Lunaophis aquaticus gen. et sp nov. is snake-like and has a combination of characters only present in these squamates, including well-developed subcentral ridges, zygosphene-zygantrum accessory articulation, well-differentiated diapophyses and parapophyses in all known vertebrae and vertebral centrum with concave ventral surface (Hoffstetter & Gasc, 1969; Estes, De Queiroz & Gauthier, 1988). The strongly posteriorly inclined neural spine in specimen MNCN-1827-E is reminiscent of the condition in lizards (Estes, De Queiroz & Gauthier, 1988). The absence of prezygapophyseal processes is characteristic of lizards but is also present in some basal snakes (Hoffstetter & Gasc, 1969).

In the last few years, efforts to understand the origins and evolution of snakes have resulted in several phylogenetic analyses that include extinct species. Crown snakes are split into two major extant clades: scolecophidians, which includes blind snakes and thread snakes, and alethinophidians, which comprises all other snakes. Nevertheless, recent analyses indicate that the unambiguously terrestrial fossil snakes Dinilysia patagonica, Najash rionegrina, and Coniophis precedens Marsh, 1892 represent more remote hierarchical sisters to crown snakes, with Dinilysia representing the immediate sister to the crown (Zaher & Scanferla, 2012; Longrich, Bhullar & Gauthier, 2012; Gauthier et al., 2012). These fossil species, in combination with the revised phylogenetic position of the limbed Tethyan marine snakes (simoliophiids) as nested within alethinophidians (Gauthier et al., 2012), contradicts the marine origin hypothesis for snakes (Hsiang et al., 2015).

The vertebrae of Lunaophis aquaticus differ significantly from extant scolecophidians in having paradiapophyses differentiated into two surfaces (diapophysis and parapophysis), the presence of a precondylar constriction, the absence of a prezygapophyseal process, and the non-oval cotyle and condyle. The former two characters also distinguish Lunaophis aquaticus from the extinct Coniophis precedens, whereas the third character-state contrasts with the condition in Dinilysia patagonica which has a prezygapophyseal process. The same combination of features observed in Lunaophis aquaticus is present in primitive snakes such as aff. Parviraptor estesi and Najash rionegrina, and at least the two latter features can be verified in Seismophis septentrionalis (Zaher, Apesteguía & Scanferla, 2009; Caldwell et al., 2015; Hsiou et al., 2014). Similar to these taxa, the posterior border of the neural arch in Lunaophis aquaticus is not notched, but it differs significantly in the position of the paradiapophyses below the vertebral centrum and the fact that these face ventrally. In the other species the paradiapophyses are more highly positioned and face more laterally than ventrally. Other differences with aff. Parviraptor estesi and Diablophis gilmorei are the better developed zygosphene, a deeper precondylar constriction, and a non-trifoliate neural canal. A neural spine reduced to a ridge is reminiscent of the condition in Coniophis precedens (but it ends in a tuberosity in that species). Lunaophis aquaticus also shares with Coniophis precedens, Dinilysia patagonica, and Seismophis septentrionalis the presence of a depressed neural arch. On either side of the neural spine, dorsal ridges are present in these snakes, as well as in Najash rionegrina and Madtsoiidae. Lunaophis aquaticus also differs from Seismophis septentrionalis in the concave zygosphene and the absence of parazygantral foramina. The vertebral centrum in Lunaophis aquaticus differs from other primitive species in not being as markedly wider anteriorly as it is in Najash rionegrina, Dinilysia patagonica, and Seismophis septentrionalis. Based on figures in Caldwell et al. (2015), a centrum that is not much wider anteriorly is present in aff. Paviraptor estesi and Diablophis gilmorei.

In comparison to other Cenomanian snakes, Lunaophis aquaticus slightly resembles simoliophiids, which represent the earliest invasion of the sea in the evolution of snakes (Hsiang et al., 2015). Simoliophiids are known from Western Europe and northwesternmost Africa to the Middle East. Among them, the better known species are Pachyrachis problematicus Haas, 1979, Haasiophis terrasanctus Tchernov et al., 2000 and Eupodophis descouensis Rage & Escuillié, 2000, whereas other species are Pachyophis woodwardi Nopcsa, 1923, Simoliophis rochebrunei Sauvage, 1880 and Simoliophis libycus Nessov, Zhegallo & Averianov, 1998 (Rage & Escuillié, 2003). Comparisons are difficult because Lunaophis is known to date only by few isolated vertebrae and a small fragment of five articulated vertebrae, whereas most simoliophiids consist of articulated skeletons (including skulls and hindlimbs), in which articular surfaces, apophyses and other details of the vertebral column are hidden. An absence of a prezygapophyseal process, strongly posteriorly inclined neural spine in anterior vertebrae, and concave zygosphene are features shared with simoliophiids but also lizards and some primitive snakes. All simoliophiids display pachyostosis that increases the mass of the vertebrae and ribs. Pachyostosis in simoliophiids species is pronounced, with the centrum and neural arch being swollen in all dimensions, especially in Pachyophis and Simoliophis (Lee & Caldwell, 1998; Houssaye, 2010; Rage, Vullo & Néraudeau, 2016). However, the pachyostosis in Lunaophis is less pronounced, and is concentrated on the prezygapophyseal bodies and on the posterior part of the neural arch, above the postzygapophyses.

The vertebrae of Simoliophis are the best described among simoliophiids, owing to the fact that they are poorly exposed and are not known three-dimensionally in other simoliophiid taxa. Rage & Escuillié (2003) and Rage, Vullo & Néraudeau (2016) consider the vertebrae of the marine hindlimbed snakes from the Cenomanian to be very similar to those of Simoliophis; more specifically, they indicate that what is known of the vertebrae of Pachyrhachis and Haasiophis do not permit any distinction between these two snakes and Simoliophis. Based on the best descriptions of Pachyrhachis problematicus and Simoliophis rochebrunei (Lee & Caldwell, 1998; Rage, Vullo & Néraudeau, 2016), mid-trunk vertebrae of these species have higher neural arches than those of Lunaophis, and vertebral centra are short and broad instead long and narrow, as in the new genus. Despite variations in height, neural spines on all precloacal vertebrae of Simoliophis and Pachyrhachis are well-developed as a distinctive process, whereas in Lunaophis the neural spines are well-developed only in anterior vertebrae, but they become greatly reduced and finally disappear in mid- and posterior precloacal vertebrae. The posterodorsal border of the neural arch is straight in Simoliophis but convex in Lunaophis. The zygosphene of Simoliophis is proportionally smaller, is very narrow and its roof appears as a high triangle, different from the wide and dorsally flat zygosphene of Lunaophis. Other distinctive characters of Simoliophis differing from those found in Lunaophis are the higher position and stronger inclination of the prezygapophyses; the interzygapopyseal constriction is shallower; the paradiapophyses are larger, high-up, face laterally and reach the level of the prezygapophyseal extremity; and the form of the centrum which does not narrow posteriorly. Aside from these characters and according to the most complete descriptions of Haasiophis (Rieppel et al., 2003), the neural spines are low throughout the precloacal region of this genus but they do not disappear as in the posterior trunk vertebrae of Lunaophis. In addition, evident pachyostosis in midtrunk vertebrae that affects the paradiapophyses is a distinctive feature of Haasiophis which is not evident in Lunaophis. Vertebrae of Eupodophis look significantly higher than in Lunaophis and have characteristic protuberances on the neural arch on either side of the neural spine, which represents a unique character of this genus (Rage & Escuillié, 2000).

In conclusion, the vertebrae described here constitute a distinctive taxon that displays features that distinguish it from other known extinct and extant snakes. Outstanding characters of this snake are the walls of the neural arch arising from the midline of the vertebrae and diverging to the subcentral ridges, the pachyostosis on the prezygapophyses and posterior neural arch, the depressed and laterally expanded roof of the neural arch with strong development of the interzygapophyseal ridges, the extension of the arch beyond the level of the condyle and forming a convex posterior edge, the long and narrow centrum with strong lateral development of the subcentral crests, the low placed and closely spaced paradiapophyses with small articular surfaces facing ventrally and projected beyond the ventral rim of the cotyle with a short process, and the substantial changes of the neural spines along the precloacal region from well-developed as a tubular process to an absolutely lacking spine. Although some features available on vertebrae of Lunaophis are plesiomorphic (anterior border of zygosphene clearly concave, articular facets of zygapophyses well inclined on horizontal, prezygapohyseal processes lacking, posterior edge of neural arch not notched, neural spine of anterior vertebrae posteriorly projected), the mentioned singular characters have not been not described for any other snake and make Lunaophis an enigmatic new taxon from the South American Cenomanian.

Mode of life of Lunaophis aquaticus. Some features of the vertebrae of Lunaophis are indicative of an aquatic mode of life. The depressed neural arches associated with the lacking or greatly reduced neural spines is a feature shared by fossorial snakes such as scolecophidians and anilioids, and other burrowing squamates such as amphisbaenians (Hoffstetter & Gasc, 1969). Nevertheless, the presence of this feature contrasts with the ventrally placed paradiapophyses, the medium size of the vertebrae, and the well-developed neural spine shown by vertebra MNCN-1827-E, which argues against possible fossorial habits. In particular, closely spaced paradiapophyses oriented in a ventral position with articular surfaces that face ventrally indicate that the ribs were directed below the vertebral centra and that the body of the snake was likely strongly compressed laterally as an adaptation for swimming. Elongate bodies of snakes are efficient for swimming, but all extant species of sea snakes have evolved paddle-like tails and many have laterally compressed bodies which give them an eel-like appearance and increase their locomotory ability in water. A laterally compressed body helped by well-developed muscles permits an efficient propulsion into the water. Thus, the body morphology of Lunaophis clearly argues for a highly aquatic mode of life. In the same sense, pachyostosis s.l., is relatively frequent in aquatic tetrapods living in shallow marine environments (e.g., Ricqlès & Buffrénil, 2001). Thus, the pachyostotic condition of studied vertebrae also supports an aquatic mode of life for Lunaophis.

Paleoenvironment and paleoecology. As discussed above, Lunaophis aquaticus gen. et sp. nov. represents an aquatic lineage of snakes that exploited marine environments. This is also reflected by the depositional conditions of the La Luna Formation, interpreted as a typical marine environment where laminated organic rich intervals suggest a deposition on the mid-shelf to upper continental slope under anoxic or poorly oxygenated conditions (Macellari & De Vries, 1987; Erlich et al., 1999; Bralower & Lorente, 2003; Zapata et al., 2003). The organic matter of the sediments in the La Aguada Member (Trujillo area) is mostly of algal origin (Tribovillard et al., 1991). González de Juana, Iturralde de Arocena & Picard (1980) suggested that the La Aguada Member could be considered as a transitional environment between the shallow waters of the Maraca formation (or La Puya Member according to Renz, 1959; Renz, 1968) and the pelagic facies of the La Luna Formation. In contrast with the pelagic and hemipelagic deep water sedimentation suggested by Tribovillard et al. (1991), Erlich et al. (1999) and Méndez (1981) suggested that the anoxic conditions of the La Luna Formation during the late Albian-early Cenomanian transgression were not due to water depth but pre-existing anoxic conditions in the slope zone. On basis of benthic and planktonic foraminiferans, Méndez (1981, and references therein) recognized an increase in the submersion of the platform, but probably with depths that did not exceed 50 m.

The holotype of Lunaophis aquaticus gen. et sp nov. is associated with other marine vertebrates (sharks and bony fishes) in the Aguada Member (Cementos Andinos quarry). Bony fish remains are very abundant in the horizon yielding Lunaophis aquaticus and adjacent strata (Fig. 2). These remains include isolated and semi-articulated cranial and postcranial elements of Xiphactinus audax (Leidy, 1870; Carrillo-Briceño, Alvarado-Ortega & Torres, 2012), ichthyodectiforms, enchodontids and small indeterminate fishes. The chondrichthyans are represented mainly by isolated teeth of at least three species of lamniform sharks although a semi-complete, articulated vertebral column of a lamniform species has also been recovered (all these specimens are currently under study). Benthic invertebrates are scarce in the shales of the Cementos Andinos quarry; however, small indeterminate bivalve molds are common in the limestones. The benthic invertebrate fauna in the La Aguada Member could represent brief periods of better oxygenated conditions on the sea floor or organisms that were tolerant to anoxic environments, as has been suggested forother sections of the La Luna Formation (e.g., Tribovillard et al., 1991). Although anoxic-dysoxic conditions prevailed on the seafloor of the basin (Méndez, 1981; Macellari & De Vries, 1987; Tribovillard et al., 1991; Erlich et al., 1999), the presence of ammonites (Renz, 1959; Renz, 1982), reptiles (Lunaophis aquaticus), and abundant fishes provides evidence of well-oxygenated surface waters, indicating that the Aguada Member environment was characterized by a stratified water column. In addition, other chondrichthyans, bony fishes and marine reptile have also been found throughout the La Luna Formation (Weiler, 1940; Moody & Maisey, 1994; Casas & Moody, 1997; Sánchez-Villagra, Brinkmann & Lozsán, 2008; Carrillo-Briceño, 2009; Carrillo-Briceño, 2012).

Conclusion

Lunaophis aquaticus gen. et sp. nov. is the oldest known snake from northern South America. It is of Cenomanian age, with relevant primitive features that distinguish it from other fossil and extant taxa by a number of characters that make it a new and enigmatic taxon, without evident affinities with any particular snake group. The nature of the fossils does not permit phylogenetic analysis until skull bones are available. The anatomical features of Lunaophis aquaticus with a laterally compressed body, the pachyostosis of the vertebrae, the depositional conditions of the La Luna Formation and the associated fauna clearly support a tropical marine paleoenvironment where this snake would have displayed an aquatic mode of life. In the context of the phylogeny proposed by Hsiang et al. (2015), Lunaophis aquaticus represents the oldest snake to have adopted an aquatic mode of life outside of the African and European Tethyan and Boreal Zones.

The authors wish to especially thank Lilia Vierma (†), Carlos Torres and Cemento Andino Ca., for their valuable assistance in the field; to Alfredo Carlini for his substantial assistance in making this collaborative work possible; to Marcelo Sánchez Villagra, Torsten M. Scheyer, Christian Kolb and members of the Evolutionary Morphology and Palaeobiology group at the Palaeontological Institute and Museum, University of Zurich, Switzerland, for generous and significant counseling and collaboration; Alexandra Wegmann for conducting the scanning; and to the Instituo del Patrimonio Cultural de Venezuela for the authorization and collecting permission. The Academic Editor Hans-Dieter Sues, and the reviewers Alexandra Houssaye and Michael Caldwell, are thanked for providing helpful comments that improved the manuscript.

Additional Information and Declarations

Competing Interests

Author Contributions

Data Availability

New Species Registration

The authors declare there are no competing interests.

Adriana Albino and James M. Neenan conceived and designed the experiments, performed the experiments, analyzed the data, contributed reagents/materials/analysis tools, wrote the paper, prepared figures and/or tables, reviewed drafts of the paper.

Jorge D. Carrillo-Briceño conceived and designed the experiments, performed the experiments, analyzed the data, contributed reagents/materials/analysis tools, wrote the paper, prepared figures and/or tables, reviewed drafts of the paper, field collection and preparation process of the fossil specimens.

The following information was supplied regarding data availability:

All illustrations and 3D models deposited at Morphobank: doi: 10.18563/m3.2.2.e2.

The following information was supplied regarding the registration of a newly described species:

Lunaophis gen. nov. LSID: http://zoobank.org/NomenclaturalActs/D269D7FA-1B3A-431B-85AF-B081E1434049;

Lunaophis aquaticus sp. nov. LSID: http://zoobank.org/NomenclaturalActs/175D3D55-D85A-4013-8D30-563BAB7A4143;

Publication LSID: urn:lsid:zoobank.org:pub:918B6879-8908-488F-876B -EA741DFF627B.

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
