# Peer review of "An enigmatic aquatic snake from the Cenomanian of Northern South America"

_PeerJ, doi:10.7717/peerj.2027_

## Round 0.1 · original submission · Major Revisions

Both reviewers have comments (and both have supplied annotated PDFs), which must be addressed carefully. The new taxon requires a proper diagnosis that really should highlight the autapomorphic features of the new taxon.

The manuscript was poorly written, and the AE has edited the manuscript (see my appended PDF). Please make sure that Dr. Neenan (a native English speaker) reviews the final version of the text.

·

Basic reporting

Dear Editor,

This manuscript describes a new fossil snake from the Cenomanian of South America. Snake material from this period is rare and of extreme importance to better understand snake origin. This paper is thus very important for the scientific community.
The paper is well written and clear. I have made various comments throughout the manuscript with suggestions as I think that the description could be clearer and, especially, more detailed and more "justified". Notably various hypotheses that are written as "solid conclusions" whereas there is no real arguments.
Also, the use of microtomography is not justified anywhere and we can wonder why it was done; however, additional data could be available thanks to the scan data and are not added in the manuscript. If possible, that would be great to add them. (see comments in the MS)
These modifications will probably be easy to do for the authors. That is why I consider this review as minor/moderate.
If my suggestions are unclear, I remain at the disposal of the authors.
Best regards,

Alexandra Houssaye

Experimental design

See comment about microtomography above

Validity of the findings

See comment above about "justification"

·

Basic reporting

The article is well written in terms of English usage, though there are a few minor editorial comments made on the manuscript.

Relevant literature is cited, but it is somewhat superficial in terms of the connections made between the new material and the existing data and hypotheses as cited. For example, the authors discuss "prescolecophidian snakes" and consider their new taxon to be one of these. However, I have no idea what a prescolecophidian snake is, can find no definition of this obvious grade of snake, and need to see that term defined in their text, or better yet, abandoned. In any current snake phylogeny, scolecophidians are nesting, right or wrong, between pachyophiids and Najash, Dinilysia, etc. The vertebrae of the new taxon are more like pachyophiids, and not at all like Najash or the madtsoiids. This needs to made clear in the text as a vague reference to "prescolecophidian" is not sufficient nor informative.

Figures are relevant and informative, though a line drawing of an isolated vertebra would not hurt.

The submission is a coherent body of work.

Experimental design

The research as described is original and has a clearly defined research question - i.e., new snake fossil and anatomy helps answer the question as to what kind of snake and how this new taxon relates to current understanding of snake evolution.

Methodology is appropriately outlined and described.

I see no breach of ethics.

Validity of the findings

The data is good, though the conclusions are not as exploratory as I would like them to be - see above criticism regarding "prescolecophidian"

Additional comments

see above. The manuscript is acceptable but needs minor revisions. I am expecting editorial corrections, a more robust discussion of phylogeny, not this weak assignment to "prescolecophidian", and some redefining of the use of the term pachyostosis and is regionalization to the neural arch, but not the centrum. Otherwise, I recommend acceptance with minor revisions.

---

## Round 0.2 · accepted · Accept

Thank you for carefully addressing the reviewers' and academic editor's comments.